# Evaluating the effects of community-based programs on viral rebound and viral suppression among HIV-positive orphaned and vulnerable children receiving antiretroviral treatment: Findings from the ACHIEVE project in Tanzania

Amon Exavery[1]*, Daisy Kisyombe[2], Alison Koler[2], John Charles[1], Nemes Mallya[1], Erica Kuhlik[2], Asheri Barankena[1], Rose Fovo[1], Godfrey Martin Mubyazi[3], Tom Ventimiglia[2], Jennifer Mulik[2], Gloria Sangiwa[2], Levina Kikoyo[1]

1 Pact Tanzania Country Office, Dar es Salaam, Tanzania, 2 Pact Inc., Washington, DC, United States, 3 National Institute for Medical Research (NIMR), Dar es Salaam, Tanzania

* aexavery@pactworld.org; aexavery@gmail.com

## Abstract

### Background

Achieving and sustaining viral load suppression (VLS) among children living with HIV remains challenging despite the availability of antiretroviral therapy (ART), primarily due to adherence difficulties. This study evaluated the effects of three interventions under the ACHIEVE project on viral load outcomes (viral rebound and viral suppression) among HIV-positive orphaned and vulnerable children aged 0–17 years (hereafter referred to as CLHIV) receiving ART in Tanzania.

### Methods

This is a longitudinal analysis of 26,257 CLHIV who are beneficiaries of the ACHIEVE project in Tanzania, each with at least two viral load (VL) test results between 2021 and 2023. VL outcomes were assessed using two analytic groups: Group 1 included 21,448 CLHIV with undetectable VL (<50 copies/mL) at baseline, where those maintaining undetectable VL were compared with those experiencing viral rebound (VL ≥ 50 copies/mL) at follow-up. Group 2 comprised 4,809 CLHIV with detectable VL (≥50 copies/mL) at baseline, where those achieving undetectable VL at follow-up were compared with those who did not. The study used a multivariable logistic regression model and propensity score matching (PSM) to evaluate the effects of ACHIEVE project interventions–WORTH Yetu economic strengthening, linkage to teen/paediatric clubs, and health insurance–on viral rebound (Group 1), and the achievement of an undetectable viral load (Group 2).

**Data availability statement:** All relevant data are within the manuscript and its Supporting information files.

**Funding:** This study draws on data from the ACHIEVE project, which was implemented by Pact in Tanzania with funding from the U.S. President's Emergency Plan for AIDS Relief (PEPFAR) through the former United States Agency for International Development (USAID) under grant number 7200AA19CA00006, awarded to Pact Inc. Authors LK, GS, AE, DK, AK, JC, NM, EK, AB, RF, and TV received salaries through their employment with Pact during the implementation of the ACHIEVE project. The funders had no role in study design, data collection and analysis, decision to publish, or preparation of the manuscript.

**Competing interests:** I have read the journal's policy and the authors of this manuscript have the following competing interests: One author, Amon Exavery, was awarded a travel scholarship by the International AIDS Society (IAS) to attend and present an abstract based on Group 2 respondents at the HIVR4P 2024, the 5th HIV Research for Prevention Conference, held in Lima, Peru, October 6–10, 2024. The rest of the authors declared that no competing interests exist.

## Findings

Overall, 13.1% (2,805/21,448) of Group 1 CLHIV experienced viral rebound at follow-up. CLHIV who received ACHIEVE interventions were 23.2% less likely to experience viral rebound compared to those who did not (aOR = 0.768, 95% CI 0.668–0.882, $p < 0.001$). In Group 2, 70.9% (3,411/4,809) achieved an undetectable viral load at follow-up, with those who received ACHIEVE interventions being 31.9% more likely to achieve an undetectable viral load compared to those who did not (aOR = 1.319, 95% CI 1.059–1.643, $p = 0.014$). These findings were confirmed in the PSM analysis (viral rebound: $\beta = -0.047$, 95% CI –0.072 – –0.021, $p < 0.001$; achieving undetectable viral load: $\beta = 0.10$, 95% CI 0.04–0.16, $p = 0.001$).

## Conclusion

The ACHIEVE project interventions were significantly associated with a reduced likelihood of viral rebound, and an increased likelihood of achieving undetectable viral load among CLHIV, highlighting their potential to enhance ART outcomes. These findings suggest that expanding similar community-based interventions could further contribute to HIV treatment efficacy and support more children in achieving and maintaining an undetectable viral load, particularly those at risk of viral rebound or persistent high viral load. Future research should explore strategies for scaling these interventions and evaluating their long-term impact.

## Introduction

Although achieving viral load suppression (VLS) remains the primary goal of human immunodeficiency virus (HIV) treatment, challenges persist in sustaining it throughout the lives of people living with HIV (PLHIV). More challenges are evident in paediatric and adolescent populations [1], emphasizing the need for more tailored care and support. Due to factors such as drug resistance and poor adherence to antiretroviral therapy (ART), virally suppressed PLHIV can experience viral rebound (i.e., viral load of 50 or more copies/mL after previously being suppressed [2]) resulting from immunological failure, consequently promoting disease progression and transmission [3], as well as morbidity and mortality [4].

Extant statistics show that up to 38.0% of children on ART experience virological failure, depending on their age and ART regimen [3]. A recent study in Tanzania estimated virological failure of 34% among children aged 1–19 years [4].

Regular monitoring of viral load has been recommended by the World Health Organization (WHO) as an essential strategy for ART success and improved health outcomes among PLHIV [5]. ART adherence challenges among children living with HIV have been well-documented [6–14]. As such, effective implementation of this recommendation necessitates an understanding of children living with HIV who are unsuppressed, and those at an elevated risk of viral rebound that should be targeted with additional care and support. Because of this, the United States President's

Emergency Plan for AIDS Relief (PEPFAR) emphasizes that clinical services provided to children living with HIV in facility settings should be complemented by high-quality community-based social support for an effective response to the AIDS epidemic [15].

While interventions with community-based services have demonstrated improvement in the retention for care among adults [16], rigorous evidence on their effects on HIV clinical outcomes among children and adolescents living with HIV is limited [17], particularly among those who are orphaned and vulnerable. However, an emerging body of research on community-based interventions, including ART-related education, psychosocial support, and economic strengthening is beginning to demonstrate promising results [18,19]. A systematic review conducted in low- and middle-income countries found that "community-based interventions improved HIV prevention and treatment outcomes (for mothers and children) compared to facility-based approaches alone" [20]. The Konga model in Tanzania was associated with viral load suppression among children living with HIV through home-based approaches that addressed ART non-adherence, including counseling, psychosocial support, and tuberculosis screening [21]. Additional studies have reported that these interventions enhance HIV awareness, promote risk reduction [22], and support ART adherence [23].

This study provides important evidence to inform strategies that promote viral suppression and mitigate viral rebound, contributing to the holistic well-being of children living with HIV. The study addresses two objectives within the context of an orphaned and vulnerable children (OVC) project in Tanzania. First, it evaluates the effects of the Adolescents and Children HIV Incidence Reduction, Empowerment, and Virus Elimination (ACHIEVE) project interventions (i.e., WORTH Yetu economic strengthening, linkage to teen/paediatric clubs, and health insurance) and other factors on viral rebound, defined as relapse from undetectable (< 50 copies/mL) to detectable (≥ 50 copies/mL) viral load, among CLHIV receiving ART who had undetectable viral load at baseline. Second, it assesses the effects of the three ACHIEVE project interventions on achieving an undetectable viral load at the follow-up among CLHIV receiving ART who had detectable viral load at baseline.

## Materials and methods

### Study design and settings

This was a prospective longitudinal cohort study, analysing baseline data (July 1, 2021–June 30, 2022) and follow-up data (July 1, 2022–July 15, 2023), for CLHIV on antiretroviral therapy (ART) who were enrolled in the ACHIEVE project in Tanzania. The data were extracted from the ACHIEVE project's databases on September 13, 2023. The ACHIEVE project is an OVC program which aimed at strengthening the systems, structures, and capacities of social service providers at the national and community levels for the delivery of quality services to OVC, at-risk adolescent girls and young women (AGYW), and PLHIV. Furthermore, the project provides family-based and child-centered services directly to OVC and their caregivers, as well as need-based referral services across several areas, including HIV, other health services, nutrition, education, social services, economic strengthening, and more. More information about the ACHIEVE project is available [24]. CLHIV from the ACHIEVE project included in this study resided in 78 district councils across 15 regions of Tanzania, namely Dar es Salaam, Geita, Kagera, Katavi, Kigoma, Mara, Mbeya, Mjini Magharibi, Mwanza, Pwani, Rukwa, Shinyanga, Simiyu, Songwe, and Tabora.

### Participants

Participants in this study were CLHIV aged 0–17 years who were receiving ART and enrolled in the ACHIEVE project. Enrollment was conducted at the household level, and ACHIEVE enrollment criteria constituted HIV-related vulnerabilities, including the presence of ≥ 1 child living with HIV in the household, ≥ 1 child born to or breastfed by a woman living with HIV, or ≥ 1 child of a female sex worker within the household, and others as further clarified elsewhere [24]. Trained Community Case Workers (CCWs) carried out enrollment at both community and facility levels. Additional clinical data on

the viral load of the CLHIV were obtained from Care and Treatment Centres (CTCs) through clinical partners collaborating with the ACHIEVE project in Tanzania. The CLHIV eligible for this study had at least two viral load test results between October 2021 and June 2023. CLHIV with one or no documented viral load test results were excluded.

Data collection involved obtaining baseline and follow-up measurements for each participant as part of routine program implementation. At baseline, 99.7% of the CLHIV were reported as adherent to ART based on self-reports or caregiver accounts (see Table 1 in S1 File). The analysis focused on two groups as presented in Fig 1:

Group 1: CLHIV who had been on ART for at least six months and had at least two clinically confirmed viral load test results spaced at least six months apart between July 2021 and July 2023. These participants had an undetectable viral load (i.e., viral load < 50 copies/mL) at the first test/baseline. Viral rebound was defined as a relapse from undetectable to detectable viral load (≥ 50 copies/mL) at the second test/follow-up (or the most recent test if more than two tests were conducted) [25].

Group 2: CLHIV who had a detectable viral load at baseline. This group was assessed to determine the extent to which they achieved viral suppression to undetectable levels at follow-up and to evaluate the effect or associations of the ACHIEVE project interventions on this outcome.

### Variables

**Outcome variables.** This study analysed two dependent variables (i.e., outcomes): viral rebound, and viral suppression to an undetectable level. According to the UNAIDS, an HIV viral load of less than 50 copies per millilitre of blood is defined as undetectable, hence untransmittable (U = U) [25]. In this study, CLHIV who had undetectable viral load at baseline (Group 1) were further tracked to examine their viral load status at the follow-up. As shown in Equation (1), viral rebound at follow-up was a binary dependent variable coded '1' if the CLHIV had viral load ≥ 50 copies/mL and '0' if the CLHIV maintained the undetectable (< 50 copies/mL) status from baseline to follow-up.

$$\text{Viral rebound at follow}-\text{up} \ = \ \begin{cases} 0, & \text{viral load} < 50 \text{ copies/mL} \\ 1, & \text{viral load} \ \geq 50 \text{ copies/mL} \end{cases}$$

(1)

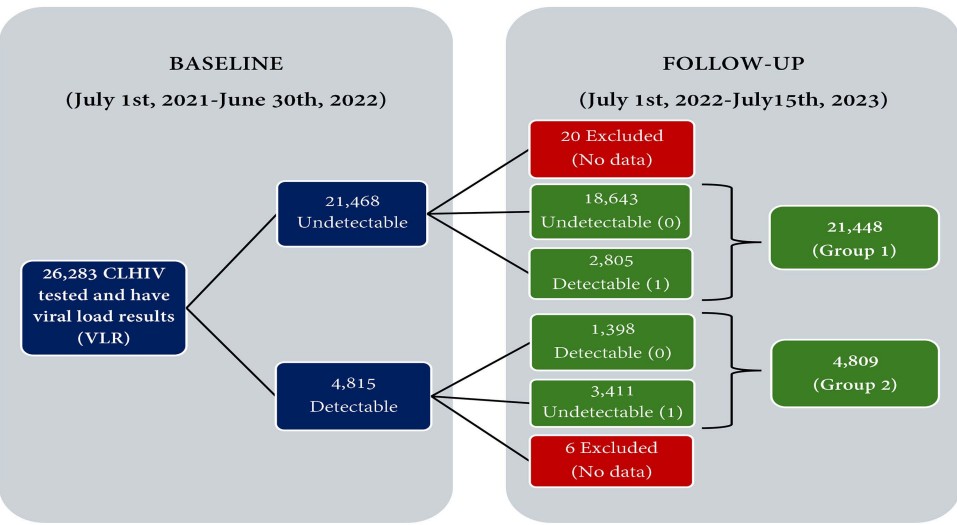

**Fig 1. Participants flow diagram at baseline and follow-up.**

Viral suppression to an undetectable level, which was assessed in Group 2 CLHIV, was also a binary dependent variable coded '1' if the CLHIV was undetectable (<50 copies/mL) at the follow-up, and '0' if the CLHIV remained detectable across the baseline and follow-up assessment periods as indicated in Equation (2).

$$\text{Undetectable viral load at follow-up} = \begin{cases} 0, & \text{if viral load} \geq 50 \text{ copies/mL} \\ 1, & \text{if viral load} < 50 \text{ copies/mL} \end{cases} \tag{2}$$

**Independent variables.** This study included several independent (explanatory) variables. The variables were divided into two sets. The first set constituted the different ACHIEVE project interventions implemented, or services provided to the CLHIV. As depicted in Table 1 in S1 File, the ACHIEVE project offered numerous services to its beneficiaries. For this study, three services were included in the analysis: 1) caregiver participation in an economic strengthening intervention using the WORTH Yetu savings and lending model, 2) provision of health insurance for the family, including the CLHIV, and 3) child linkage in teen/paediatric adherence clubs. The services were selected based on the criteria that all CLHIV were eligible to benefit, either directly or indirectly through their caregivers, and that the coverage of each service as of July 15th, 2023, was not below 5% and did not exceed 95%. Services with coverage of less than 5% or more than 95% as of July 15th, 2023, were excluded because they were either commonly available or unavailable to everyone. This was necessary to ensure that there were statistically enough CLHIV who had or had not received the services for meaningful comparisons in the statistical analyses. Services provided to a subset of CLHIV due to specialized needs (e.g., transport assistance for CLHIV facing adherence issues) were also excluded from the analysis.

The ACHIEVE project contributed to each of the three services in different ways. Regarding WORTH Yetu groups, in which the project implements an enhanced village savings and loans group model, ACHIEVE was fully responsible for establishing and managing them. The intervention aimed to strengthen the caregivers' economic capacity to meet the needs of OVC and their own household needs by improving livelihoods and financial resilience. It was implemented through locally formed groups (i.e., WORTH Yetu groups) with 15–30 members each that met weekly, where members made mandatory and voluntary savings to build capital for individual loans and group start-up activities, alongside access to financial literacy, savings opportunities, and microcredits from financial institutions and other sources to support their income-generating activities such as farming, animal husbandry, and horticulture. More details about the WORTH Yetu groups are published elsewhere [26–29].

Teen/paediatric clubs operate within health facilities and serve as platforms for peer and clinical support to promote ART adherence among children and adolescents living with HIV. In Tanzania, these clubs are organized into age-specific groups to ensure appropriate engagement in line with the national HIV service delivery approaches, with distinct groups for children (0–14 years), adolescents (15–17 years), and adults (18 years and above). The clubs complement routine clinical care by offering structured peer engagement and age-specific guidance related to treatment adherence, HIV status disclosure, positive living, and life skills that support overall health and well-being [15]. Within this framework, the ACHIEVE project linked CLHIV to these existing clubs and strengthened follow-up and monitoring through CCWs. The CCWs facilitated and monitored CLHIV attendance at club sessions, escorted the CLHIV for their viral load testing, ART refills, and supported participation in Enhanced Adherence Counseling (EAC) sessions as needed. Further information on these clubs and services is available elsewhere [30,31].

Concerning health insurance, the ACHIEVE project procured health insurance coverage for some families that were previously uninsured. In this case, the health insurance procured for the families was the improved Community Health Fund (iCHF). More information about the iCHF in Tanzania is available in several sources [32,33].

The second set of independent variables constituted other baseline factors that were not considered project interventions. This included the duration (in months) for which the CLHIV had been receiving services from the ACHIEVE project, the type of ART regimen the CLHIV was on at baseline (categorized as dolutegravir (DTG)-based, and others), and whether the CLHIV had changed regimen in the last 6 months. Baseline sociodemographic characteristics of the CLHIV and their caregivers' were also included: CLHIV sex, age, and school attendance status; CLHIV's caregiver sex, age, education, place of residence, family size, and household hunger. The variables and their respective categories are detailed in Table 2.

Household hunger was measured using the FANTA Household Hunger Scale (HHS) which groups households into three categories that represent their hunger severity as: (1) little to no hunger households (i.e., food secure), (2) moderate hunger households, and (3) severe hunger households [34]. Specific details of how the HHS was operationalised within the ACHIEVE project and the questions involved in the process have been described elsewhere [24].

**Statistical analysis.** Both descriptive and inferential statistics were computed in the data analysis process. In the descriptive analysis part, the frequency distribution of the respondents across each of the variables was presented. Then each of the outcomes was cross-tabulated against each of the independent variables with a Chi-square test (where applicable) to obtain bivariate associations. Finally, we conducted multivariate analysis using a logistic regression model to identify factors associated with each outcome. In each model, variables constituting the project interventions were retained in the model regardless of their influence on the outcomes. Other variables were selected for inclusion in the multivariate models if it was statistically significant, through log-likelihood ratio (LR) test, that their presence improved the overall model.

The ACHIEVE project interventions' effect on viral rebound and undetectable viral load was assessed in three scenarios described in Table 1.

Beyond examining associations, we conducted PSM to estimate the impact of the ACHIEVE project interventions (treatment) on both outcomes – viral rebound and undetectable viral load among the CLHIV. To account for selection bias or confounding effect, matching was done on all the described independent variables [35–38].

The data analysis for this study was carried out using Stata (Version 17.0) statistical software, with a 5% significance level ($\alpha = 0.05$).

**Ethics approval and consent to participate.** Ethics approval for this study was obtained from the Medical Research Coordinating Committee (MRCC) of the National Institute for Medical Research (NIMR) in Tanzania (NIMR/HQ/R.8a/ Vol.IX/4080). Written informed consent was obtained from all caregivers before their households were enrolled in the ACHIEVE project. This process ensured that caregivers were fully informed about their roles and responsibilities in the project. The consent provided by the caregivers (i.e., parents or guardians) also covered the participation of OVC under the age of 18 years in their household.

## Results

### Characteristics of respondents

As presented in Table 2, Group 1 of respondents analyzed included 21,448 CLHIV (52.4% female) aged 10.8 (±4.1) years on average who had undetectable viral load (<50 copies/mL) at baseline. Majority of the children (81.5%) were attending

**Table 1. Data analysis scenarios for assessing associations with and impact of the ACHIEVE project interventions on viral rebound and undetectable viral load among CLHIV at follow-up.**

| Scenario | Description |
|---|---|
| Scenario 1 | Each of the three interventions was included in both models as a separate variable, representing whether each child had received the intervention or not at the time of the follow-up survey. |
| Scenario 2 | A new variable constituting a row sum across the three interventions was generated to establish the number of project interventions that each child had received between the baseline and follow-up. The new variable, the number of project interventions received, had values ranging from zero (0) for CLHIV who had not received any of the three interventions to three (3) for those who had received all the three project interventions. The variable was then included in the models as a categorical variable, with CLHIV who had not received any of the three interventions (i.e., 0) constituting a reference category. |
| Scenario 3 | A new binary variable from the interventions was generated, with CLHIV who had received at least one (≥ 1) of the interventions coded '1' and those who had received none (i.e., zero) of the interventions coded '0'. This variable was used in both models to assess how viral rebound and undetectable viral load compared among CLHIV who had received at least one intervention and those who had received none.<br>This variable was also used in the Propensity Score Matching (PSM) analysis to explore if the observed associations were causal. |

**Table 2.  Baseline characteristics of respondents for both Group 1 and Group 2.**

| | Group 1: CLHIV with undetectable viral load at baseline (n = 21,448) | | Group 2: CLHIV with detectable viral load at baseline (n = 4,809) | |
|---|---|---|---|---|
| **ART regimen** | | | | |
| DTG-based | 20,858 | (97.3%) | 4,596 | (95.6%) |
| Other regimens | 590 | (2.8%) | 213 | (4.4%) |
| **Duration in the ACHIEVE project** | | | | |
| <6 months | 44 | (0.2%) | 14 | (0.3%) |
| 6-11 months | 2,558 | (11.9%) | 423 | (8.8%) |
| 12+months | 18,846 | (87.9%) | 4,372 | (90.9%) |
| **CLHIV's caregiver participates in WORTH Yetu** | | | | |
| No | 17,315 | (80.7%) | 3,891 | (80.9%) |
| Yes | 4,133 | (19.3%) | 918 | (19.1%) |
| **CLHIV linked to teen/paediatric clubs** | | | | |
| No | 2,700 | (12.6%) | 581 | (12.1%) |
| Yes | 18,748 | (87.4%) | 4,228 | (87.9%) |
| **Health insurance (iCHF)** | | | | |
| No | 15,744 | (73.4%) | 3,652 | (75.9%) |
| Yes | 5,704 | (26.6%) | 1157 | (24.1%) |
| **CLHIV sex** | | | | |
| Female | 11,241 | (52.4%) | 2,394 | (49.8%) |
| Male | 10,207 | (47.6%) | 2,415 | (50.2%) |
| **CLHIV age** | | | | |
| <5 years | 1,549 | (7.2%) | 535 | (11.1%) |
| 5-9 years | 6,695 | (31.2%) | 1,512 | (31.4%) |
| 10-14 years | 8,158 | (38.0%) | 1,719 | (35.8%) |
| 15-17 years | 5,046 | (23.5%) | 1,043 | (21.7%) |
| **CLHIV's caregiver age** | | | | |
| 18-29 years | 9,160 | (42.7%) | 2,209 | (45.9%) |
| 30-39 years | 4,120 | (19.2%) | 897 | (18.7%) |
| 40-49 years | 4,764 | (22.2%) | 1,030 | (21.4%) |
| 50-59 years | 2,025 | (9.4%) | 407 | (8.5%) |
| 60+years | 1,379 | (6.4%) | 266 | (5.5%) |
| **CLHIV school attendance status** | | | | |
| Not attending | 3,962 | (18.5%) | 1,107 | (23.0%) |
| Attending | 17,486 | (81.5%) | 3,702 | (77.0%) |
| **Level of household hunger** | | | | |
| Little to no hunger | 3,012 | (14.0%) | 680 | (14.1%) |
| Moderate hunger | 17,717 | (82.6%) | 3,955 | (82.2%) |
| Severe hunger | 719 | (3.4%) | 174 | (3.6%) |
| **Place of residence** | | | | |
| Rural | 14,325 | (66.8%) | 2,986 | (62.1%) |
| Urban | 7,123 | (33.2%) | 1,823 | (37.9%) |
| **Family size** | | | | |
| 2-3 people | 14,978 | (69.8%) | 3,374 | (70.2%) |
| 4-6 people | 6,108 | (28.5%) | 1,358 | (28.2%) |
| 7+people | 362 | (1.7%) | 77 | (1.6%) |

*(Continued)*

| | Group 1: CLHIV with undetectable viral load at baseline (n = 21,448) | | Group 2: CLHIV with detectable viral load at baseline (n = 4,809) | |
|---|---|---|---|---|
| **Caregiver sex** | | | | |
| Female | 13,957 | (65.1%) | 3,073 | (63.9%) |
| Male | 7,491 | (34.9%) | 1,736 | (36.1%) |
| **Caregiver education** | | | | |
| Never attended | 3,660 | (17.1%) | 672 | (14.0%) |
| Only primary | 16,970 | (79.1%) | 3,930 | (81.7%) |
| Secondary or more | 818 | (3.8%) | 207 | (4.3%) |
| **ART change in the last 6 months** | | | | |
| No | 18,103 | (84.4%) | 4,088 | (85.0%) |
| Yes | 3,345 | (15.6%) | 721 | (15.0%) |

school. Nearly all children (97.3%) were on DTG-based regimens. Two-thirds (66.8%) of the children resided in rural areas. The majority (87.9%) had been in the ACHIEVE project for one or more years. Regarding the interventions, 19.3% of the children had caregivers who were participants in the project's economic strengthening intervention through WORTH Yetu groups, 87.4% of the children were linked to teen/paediatric clubs, and 26.6% of the children had been supported by the ACHIEVE project to receive health insurance. The rest of the characteristics of the children are presented in Table 2.

Table 2 also presents characteristics Group 2 of children analyzed that included 4,809 CLHIV (49.8% female) aged 10.3 (±4.3) years on average who had detectable viral load (≥ 50 copies/mL) at baseline. The majority (77.0%) of this group's children attended school. Nearly all children (95.6%) were on DTG-based regimens. Many of the children (62.1%) were living in rural areas and the rest in urban areas. Nearly all of the children (90.9%) had been in the ACHIEVE project for one or more years. Regarding the project interventions, 19.1% of the CLHIV had caregivers who were participants in WORTH Yetu, 87.9% of the children were linked to teen/paediatric clubs, and 24.1% had received health insurance from the project. Further characteristics of children are presented in Table 2.

## Number of project services received

At the time of the follow-up, over 91% of the CLHIV in both groups had received at least one of the three ACHIEVE project services analysed by this study. As shown in Fig 2, the number of project services received by the CLHIV was comparable

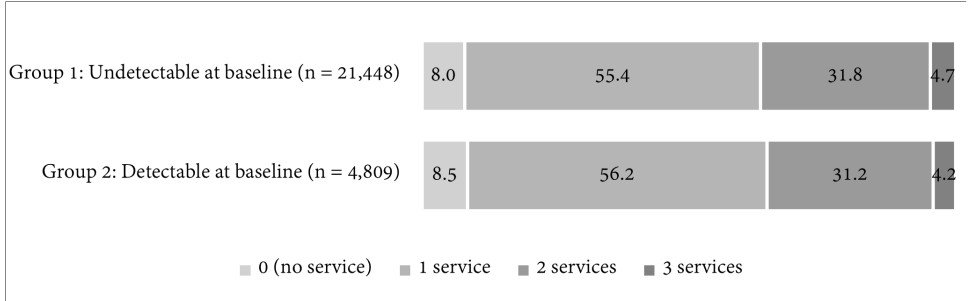

**Fig 2. Percentage of CLHIV by the number of ACHIEVE project interventions received (i.e., WORTH Yetu economic strengthening, linkage to teen/paediatric clubs, and health insurance) as of July 15th, 2023.**

in both groups. The majority (over 55%) had received one of the services, more than 31% had received any two of the services, and around 4% had received all three project services.

### Findings from the bivariate analysis

Figures 3–4, and Table 3 show the results from the bivariate analysis for both outcomes. For viral rebound (Group 1), findings revealed that 86.9% (n = 18,643) of the CLHIV maintained their undetectable status at the follow-up (second test), and 13.1% (n = 2,805) who were initially undetectable experienced viral rebound at the follow-up. The occurrence of viral rebound by the number of ACHIEVE project interventions received is presented in Fig 3 and shows a significantly declining trend ($p < 0.001$), with viral rebound ranging from 15.8% for CLHIV who received none of the three interventions to 11.0% for those who received all three interventions.

Also, viral rebound was lower among CLHIV who received each of the specific interventions, with significant reductions for participation in teen/paediatric clubs ($p = 0.015$) and having health insurance ($p < 0.001$), while a similar but non-significant pattern was observed for WORTH Yetu ($p = 0.250$) (Table 3).

Other factors (Table 3) which were significantly associated with viral rebound were ART regimen ($p = 0.001$), caregiver age ($p = 0.011$), school attendance status ($p = 0.003$), place of residence ($p < 0.001$), family size ($p = 0.002$), and caregiver education ($p < 0.001$).

The percentage of CLHIV whose viral load status changed from detectable at baseline to undetectable at follow-up (Group 2) by the number of project interventions received is presented in Fig 4, and the percentage who achieved

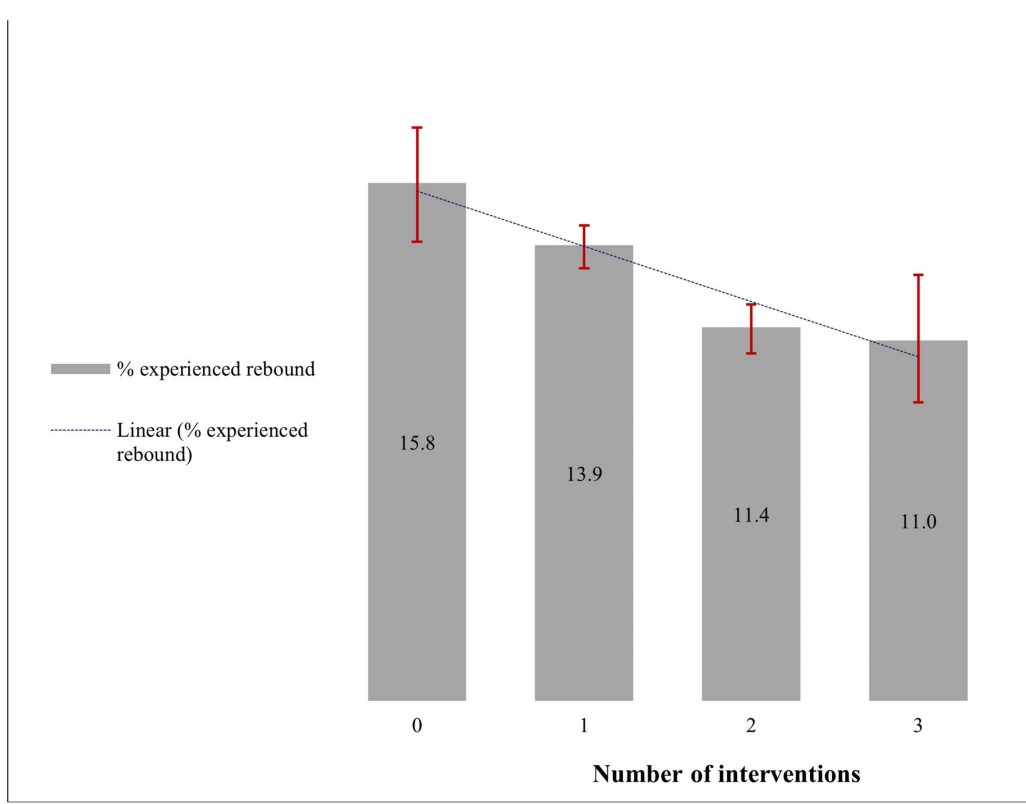

**Fig 3. Percentage of CLHIV with undetectable viral load at baseline who experienced viral rebound at the follow-up by the number of ACHIEVE project interventions received (n = 21,448). (Error bars represent 95% confidence intervals).**

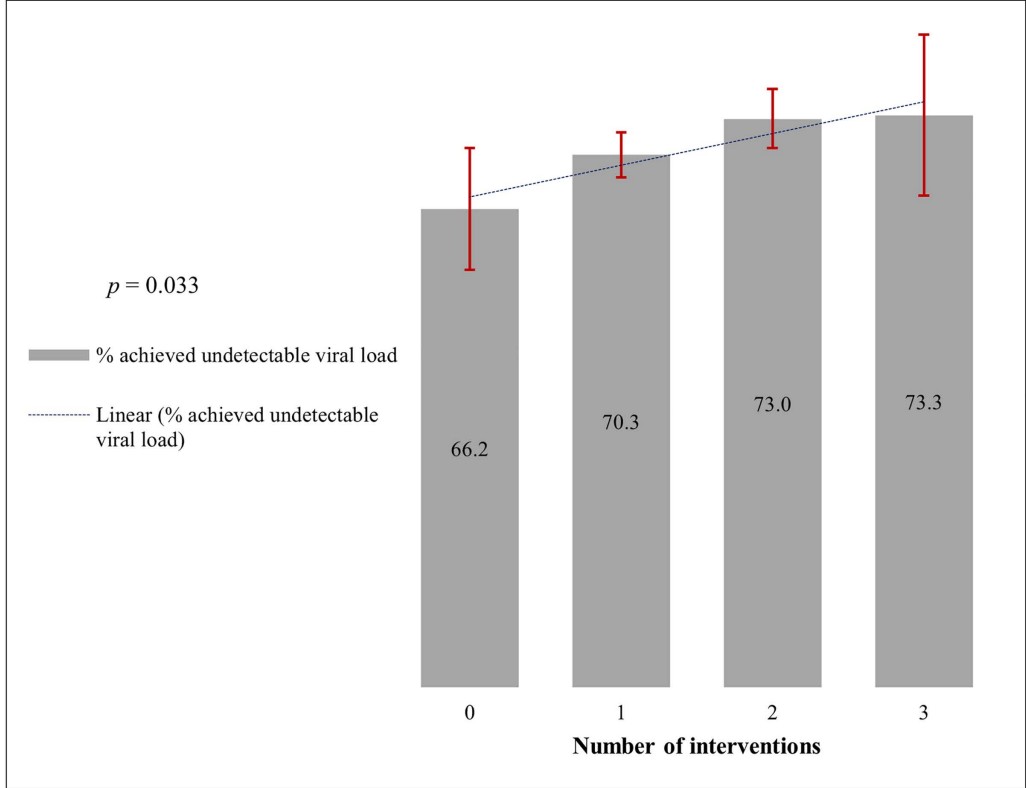

**Fig 4. Percentage of CLHIV with detectable viral load at baseline who achieved undetectable levels at follow-up by the number of ACHIEVE project interventions received (n = 4,809).(Error bars represent 95% confidence intervals).**

undetectable viral load by specific interventions is shown in Table 3. Findings revealed that 70.9% (n = 3,410) of the CLHIV with detectable viral load at baseline had achieved an undetectable status at follow-up, while the other 29.1% (n = 1,399) remained detectable at follow-up (Table 3). Moreover, the analysis identified a statistically significant trend, showing that the greater the number of ACHIEVE project interventions received, the higher the percentage of CLHIV who achieved an undetectable viral load ($p = 0.005$) (Fig 4).

Additionally, achieving an undetectable viral load was significantly higher among CLHIV who were linked to teen/paediatric adherence clubs ($p = 0.031$), or health insurance intervention ($p = 0.001$) compared to those who received no intervention, though this did not hold for WORTH Yetu ($p = 0.62$) (Table 3). Other factors associated with undetectable viral load were caregiver age ($p = 0.005$), school attendance status ($p = 0.030$), and place of residence ($p < 0.001$) (Table 3).

### Results from logistic regression analysis

Results from the multivariable logistic regression analyses showing the number of the ACHIEVE project interventions and other factors associated with both viral rebound and achieving undetectable viral load among the CLHIV at the follow-up are presented in Table 3 and Table 4, respectively. Additional logistic regression models considering the interventions as separate variables are presented in Table 2 in S1 File and Table 3 in S1 File; and models considering the interventions as a binary variable representing whether the CLHIV had received at least one intervention, or none, are presented in Table 4 in S1 File and Table 5 in S1 File.

**Table 3. Bivariate analysis of factors associated with viral rebound, and undetectable viral load among CLHIV.**

| | Group 1: Undetectable viral load at baseline (n = 21,448) | | Group 2: Detectable viral load at baseline (n = 4,809) | |
| --- | --- | --- | --- | --- |
| | % Undetectable at follow-up (n = 18,643) | % Detectable at follow-up (REBOUND) (n = 2,805) | % Undetectable at follow-up (n = 3,411) | % Detectable at follow-up (n = 1,398) |
| **OVERALL** | 86.9 (86.5–87.4) | 13.1 (12.6–13.5) | 70.9 (69.6–72.2) | 29.1 (27.8–30.4) |
| **ART regimen (1\*\*)** | | | | |
| DTG-based | 87.1 (86.7–87.5) | 12.9 (12.5–13.4) | 71.1 (69.8–72.4) | 28.9 (27.6–30.2) |
| Other regimens | 82.2 (79.1–85.3) | 17.8 (14.7–20.9) | 67.6 (61.3–73.9) | 32.4 (26.1–38.7) |
| **Duration in the ACHIEVE project** | | | | |
| <6 months | 79.6 (67.6–91.5) | 20.5 (08.5–32.3) | 57.1 (31.2–83.1) | 42.9 (16.9–68.8) |
| 6-11 months | 86.3 (85.0–87.7) | 13.7 (12.3–15.0) | 74.9 (70.8–79.1) | 25.1 (20.9–29.2) |
| 12 + months | 87.0 (86.5–87.5) | 13.0 (12.5–13.5) | 70.6 (28.1–30.8) | 29.4 (28.1–30.8) |
| **CLHIV's caregiver participates in WORTH Yetu** | | | | |
| No | 86.8 (86.3–87.3) | 13.2 (12.7–13.7) | 71.1 (69.7–72.5) | 28.9 (27.5–30.3) |
| Yes | 87.5 (86.5–88.5) | 12.5 (11.5–13.5) | 70.3 (67.3–73.2) | 29.7 (26.9–32.7) |
| **CLHIV linked to teen/paediatric clubs (1\*\*, 2\*\*)** | | | | |
| No | 85.4 (84.1–86.8) | 14.6 (13.2–15.9) | 67.1 (63.3–70.9) | 32.9 (29.1–36.7) |
| Yes | 87.1 (86.7–87.6) | 12.9 (12.4–13.3) | 71.5 (70.1–72.8) | 28.6 (27.2–29.9) |
| **Health insurance (iCHF) (1\*\*\*, 2\*\*\*)** | | | | |
| No (uninsured) | 86.0 (85.5–86.6) | 14.0 (13.4–14.5) | 69.7 (68.2–71.2) | 30.3 (28.8–31.8) |
| Yes (insured) | 89.4 (88.6–90.2) | 10.6 (09.8–11.4) | 74.8 (72.3–77.3) | 25.2 (22.7–27.7) |
| **CLHIV sex** | | | | |
| Female | 86.8 (86.1–87.4) | 13.2 (12.6–13.9) | 71.4 (69.6–73.2) | 28.6 (26.8–30.4) |
| Male | 87.1 (86.4–87.8) | 12.9 (12.2–13.6) | 70.4 (68.6–72.3) | 29.6 (27.7–31.4) |
| **CLHIV age** | | | | |
| <5 years | 88.4 (86.8–90.0) | 11.6 (10.0–13.2) | 72.3 (68.5–76.1) | 27.7 (23.9–31.5) |
| 5-9 years | 87.5 (86.7–88.2) | 12.6 (11.8–13.3) | 72.6 (70.4–74.9) | 27.4 (25.1–29.6) |
| 10-14 years | 86.3 (85.6–87.1) | 13.7 (12.9–14.4) | 70.9 (68.8–73.1) | 29.1 (26.9–31.2) |
| 15-17 years | 86.7 (85.8–87.7) | 13.3 (12.3–14.2) | 67.8 (65.0–70.6) | 32.2 (29.4–35.0) |
| **CLHIV's caregiver age (1\*\*, 2\*\*)** | | | | |
| 18-29 years | 86.1 (85.4–86.8) | 13.9 (13.2–14.6) | 69.1 (67.2–71.1) | 30.9 (28.9–32.8) |
| 30-39 years | 87.5 (86.4–88.5) | 12.6 (11.5–13.6) | 69.7 (66.7–72.7) | 30.3 (27.3–33.3) |
| 40-49 years | 87.2 (86.2–88.1) | 12.8 (11.9–13.8) | 72.8 (70.1–75.5) | 27.2 (24.5–29.9) |
| 50-59 years | 88.3 (86.8–89.7) | 11.8 (10.3–13.2) | 77.4 (73.3–81.5) | 22.6 (18.5–26.7) |
| 60 + years | 88.1 (86.4–89.8) | 11.9 (10.2–13.6) | 72.9 (67.6–78.3) | 27.1 (21.7–32.4) |
| **CLHIV school attendance status (1\*\*, 2\*\*)** | | | | |
| Not attending | 88.3 (87.3–89.3) | 11.7 (10.7–12.7) | 73.5 (70.9–76.1) | 26.5 (23.9–29.1) |
| Attending | 86.6 (86.1–87.1) | 13.4 (12.9–13.9) | 70.2 (68.7–71.6) | 29.9 (28.4–31.3) |
| **Level of household hunger** | | | | |
| Little to no hunger | 87.5 (86.3–88.7) | 12.5 (11.3–13.7) | 70.0 (66.6–73.4) | 30.0 (26.6–33.4) |
| Moderate hunger | 86.8 (86.3–87.3) | 13.2 (12.7–13.7) | 71.1 (69.6–72.5) | 29.0 (27.5–30.4) |
| Severe hunger | 86.4 (83.9–88.9) | 13.6 (11.1–16.1) | 71.8 (65.2–78.5) | 28.2 (21.5–34.8) |
| **Place of residence (1\*\*\*, 2\*\*\*)** | | | | |
| Rural | 87.8 (87.3–88.4) | 12.2 (11.6–12.7) | 72.7 (71.1–74.3) | 27.3 (25.7–28.9) |
| Urban | 85.1 (84.3–85.9) | 14.9 (14.1–15.7) | 68.0 (65.8–70.1) | 32.0 (29.9–34.2) |
| **Family size (1\*\*)** | | | | |

*(Continued)*

**Table 3.** (Continued)

| | Group 1: Undetectable viral load at baseline (n=21,448) | | Group 2: Detectable viral load at baseline (n=4,809) | |
|---|---|---|---|---|
| | % Undetectable at follow-up (n=18,643) | % Detectable at follow-up (REBOUND) (n=2,805) | % Undetectable at follow-up (n=3,411) | % Detectable at follow-up (n=1,398) |
| 2-3 people | 86.4 (85.8–86.9) | 13.6 (13.1–14.2) | 70.7 (69.1–72.2) | 29.3 (27.8–30.9) |
| 4-6 people | 88.2 (87.3–89.0) | 11.8 (11.0–12.7) | 71.5 (69.1–73.9) | 28.5 (26.1–30.9) |
| 7+people | 88.1 (84.8–91.5) | 11.9 (08.5–15.2) | 72.7 (62.8–82.7) | 27.3 (17.3–37.2) |
| **Caregiver sex** | | | | |
| Female | 86.9 (86.3–87.4) | 13.2 (12.6–13.7) | 70.7 (69.1–72.3) | 29.3 (27.7–30.9) |
| Male | 87.1 (86.3–87.8) | 12.9 (12.2–13.7) | 71.4 (69.2–73.5) | 28.6 (26.5–30.8) |
| **Caregiver education (1\*\*\*)** | | | | |
| Never attended | 89.2 (88.2–90.2) | 10.8 (09.8–11.8) | 73.7 (70.3–77.0) | 26.3 (23.0–29.7) |
| Primary | 86.6 (86.0–87.1) | 13.4 (12.9–14.0) | 70.4 (69.0–71.9) | 29.6 (28.1–31.0) |
| Secondary+ | 84.4 (81.9–86.8) | 15.7 (13.2–18.1) | 71.5 (65.4–77.6) | 28.5 (22.4–34.6) |
| **ART change in the last 6 months** | | | | |
| No | 87.1 (86.6–87.5) | 13.0 (12.5–13.4) | 70.8 (69.4–72.2) | 29.2 (27.8–30.6) |
| Yes | 86.2 (85.1–87.4) | 13.8 (12.6–14.9) | 71.6 (68.3–74.9) | 28.4 (25.1–31.7) |

1\*\*\*, 2\*\*\*: significant at p<0.0001 for Group 1, or Group 2; 1\*\*, 2\*\*: significant at p<0.05 for Group 1, or Group 2

**Table 4. Factors associated with viral rebound at follow-up among 21,448 CLHIV who had undetectable viral load at baseline in Tanzania.**

| | adjusted Odds Ratio (aOR) | Lower 95% confidence limit | Upper 95% confidence limit | *p*-value |
|---|---|---|---|---|
| **ART regimen type** | | | | |
| DTG-based | 1.000 | — | — | — |
| Other regimens | 1.377 | 1.109 | 1.711 | 0.004 |
| **Duration in the ACHIEVE project** | | | | |
| <6 months | 1.000 | — | — | — |
| 6-11 months | 0.781 | 0.370 | 1.648 | 0.520 |
| 12-23 months | 0.710 | 0.339 | 1.490 | 0.365 |
| **Number of ACHIEVE project interventions received** | | | | |
| Zero | 1.000 | — | — | — |
| One | 0.826 | 0.717 | 0.951 | 0.008 |
| Two | 0.674 | 0.579 | 0.785 | < 0.001 |
| Three | 0.664 | 0.524 | 0.843 | 0.001 |
| **CLHIV sex** | | | | |
| Female | 1.000 | — | — | — |
| Male | 0.972 | 0.897 | 1.052 | 0.480 |
| **CLHIV age** | | | | |
| <5 years | 1.000 | — | — | — |
| 5-9 years | 1.002 | 0.821 | 1.223 | 0.983 |
| 10-14 years | 1.064 | 0.856 | 1.322 | 0.576 |
| 15-17 years | 1.017 | 0.816 | 1.268 | 0.881 |
| **Caregiver age** | | | | |
| 18-29 years | 1.000 | — | — | — |
| 30-39 years | 0.862 | 0.769 | 0.966 | 0.011 |

*(Continued)*

**Table 4.** (Continued)

| | adjusted Odds Ratio (aOR) | Lower 95% confidence limit | Upper 95% confidence limit | *p*-value |
|---|---|---|---|---|
| 40-49 years | 0.876 | 0.787 | 0.976 | 0.016 |
| 50-59 years | 0.813 | 0.699 | 0.945 | 0.007 |
| 60 + years | 0.866 | 0.725 | 1.034 | 0.111 |
| **CLHIV school attendance status** | | | | |
| Not attending | 1.000 | — | — | — |
| Attending school | 1.117 | 0.970 | 1.286 | 0.124 |
| **Level of household hunger** | | | | |
| Little to no hunger | 1.000 | — | — | — |
| Moderate hunger | 1.053 | 0.936 | 1.184 | 0.393 |
| Severe hunger | 1.072 | 0.842 | 1.364 | 0.573 |
| **Place of residence** | | | | |
| Rural | 1.000 | — | — | — |
| Urban | 1.235 | 1.130 | 1.348 | < 0.001 |
| **Family size** | | | | |
| 2-3 people | 1.000 | — | — | — |
| 4-6 people | 0.883 | 0.805 | 0.969 | 0.008 |
| 7 + people | 0.909 | 0.658 | 1.257 | 0.570 |
| **Caregiver sex** | | | | |
| Female | 1.000 | — | — | — |
| Male | 1.005 | 0.923 | 1.095 | 0.910 |
| **Caregiver education** | | | | |
| Never attended | 1.000 | — | — | — |
| Primary | 1.196 | 1.066 | 1.343 | 0.002 |
| Secondary+ | 1.331 | 1.068 | 1.660 | 0.011 |
| **ART change in the last 6 months** | | | | |
| No | 1.000 | — | — | — |
| Yes | 1.158 | 1.036 | 1.294 | 0.010 |
| Constant | 0.199 | 0.092 | 0.433 | < 0.001 |

**Viral rebound.** As seen in Table 4, viral rebound (Group 1) was significantly less likely to occur in CLHIV who had received at least one of the three ACHIEVE project interventions compared to those who had received none (one vs zero: aOR = 0.826, 95% CI 0.717–0.951, *p* = 0.008; two vs. zero: aOR = 0.670, 95% CI 0.579–0.785, *p* < 0.001; three vs. zero: aOR = 0.660, 95% CI 0.524–0.843, *p* = 0.001). These results, based on the aOR, indicate that receiving all three interventions reduced the risk of viral rebound more effectively than receiving two interventions, two reduced the risk more than one, and one reduced the risk more than none.

Viral rebound was also less likely among CLHIV with older caregivers as well as those in bigger families of four or more people. On the other hand, the likelihood of viral rebound was higher among CLHIV on non DTG-based regimens (aOR = 1.377, 95% CI 1.109–1.711, *p* = 0.004), residing in urban areas (aOR = 1.235, 95% CI 1.130–1.348, *p* < 0.001), had caregivers with primary education (aOR = 1.196, 95% CI 1.066–1.343, *p* = 0.002) and secondary education or more (aOR = 1.331, 95% CI 1.068–1.660, *p* = 0.011); and had changed ART regimen in the last six months (aOR = 1.158, 95% CI 1.036–1.294, *p* = 0.01).

**Undetectable viral load.** As shown in Table 5, factors positively associated with achieving viral suppression to an undetectable level at the follow-up included the number of project interventions received amongst the three (one vs.

**Table 5. Factors associated with undetectable viral at the follow-up among 4,809 CLHIV who had detectable viral load at baseline in Tanzania.**

| | adjusted Odds Ratio (aOR) | Lower 95% confidence limit | Upper 95% confidence limit | *p*-value |
|---|---|---|---|---|
| **ART regimen type** | | | | |
| DTG-based | 1.000 | — | — | — |
| Other regimens | 0.924 | 0.684 | 1.248 | 0.610 |
| **Duration in the ACHIEVE project** | | | | |
| <6 months | 1.000 | — | — | — |
| 6-11 months | 1.746 | 0.583 | 5.230 | 0.320 |
| 12-23 months | 1.473 | 0.500 | 4.334 | 0.480 |
| **Number of ACHIEVE project interventions received** | | | | |
| Zero | 1.000 | — | — | — |
| One | 1.259 | 1.005 | 1.577 | 0.045 |
| Two | 1.432 | 1.127 | 1.821 | 0.003 |
| Three | 1.413 | 0.967 | 2.062 | 0.074 |
| **CLHIV sex** | | | | |
| Female | 1.000 | — | — | — |
| Male | 0.951 | 0.839 | 1.079 | 0.440 |
| **CLHIV age** | | | | |
| <5 years | 1.000 | — | — | — |
| 5-9 years | 1.162 | 0.887 | 1.523 | 0.280 |
| 10-14 years | 1.139 | 0.838 | 1.546 | 0.410 |
| 15-17 years | 0.988 | 0.723 | 1.351 | 0.940 |
| **Caregiver age** | | | | |
| 18-29 years | 1.000 | — | — | — |
| 30-39 years | 1.005 | 0.843 | 1.198 | 0.960 |
| 40-49 years | 1.193 | 1.007 | 1.415 | 0.042 |
| 50-59 years | 1.541 | 1.195 | 1.986 | 0.001 |
| 60＋years | 1.170 | 0.874 | 1.565 | 0.290 |
| **CLHIV school attendance status** | | | | |
| Not attending | 1.000 | — | — | — |
| Attending school | 0.820 | 0.659 | 1.021 | 0.076 |
| **Level of household hunger** | | | | |
| Little to no hunger | 1.000 | — | — | — |
| Moderate hunger | 1.066 | 0.890 | 1.276 | 0.490 |
| Severe hunger | 1.159 | 0.798 | 1.683 | 0.440 |
| **Place of residence** | | | | |
| Rural | 1.000 | — | — | — |
| Urban | 0.801 | 0.698 | 0.920 | 0.002 |
| **Family size** | | | | |
| 2-3 people | 1.000 | — | — | — |
| 4-6 people | 0.991 | 0.860 | 1.142 | 0.900 |
| 7＋people | 1.080 | 0.648 | 1.802 | 0.770 |
| **Caregiver sex** | | | | |
| Female | 1.000 | — | — | — |
| Male | 0.977 | 0.855 | 1.117 | 0.740 |
| **Caregiver education** | | | | |
| Never attended | 1.000 | — | — | — |
| Primary | 0.926 | 0.766 | 1.119 | 0.430 |

*(Continued)*

**Table 5.** (Continued)

|  | adjusted Odds Ratio (aOR) | Lower 95% confidence limit | Upper 95% confidence limit | *p*-value |
|---|---|---|---|---|
| Secondary+ | 1.080 | 0.755 | 1.547 | 0.670 |
| **ART change in the last 6 months** | | | | |
| No | 1.000 | — | — | — |
| Yes | 0.950 | 0.793 | 1.138 | 0.580 |
| Constant | 1.427 | 0.461 | 4.415 | 0.540 |

zero: aOR = 1.259, 95% CI 1.005–1.577, *p* = 0.045; two vs. zero: aOR = 1.432, 95% CI 1.127–1.821, *p* = 0.003; and three vs. zero: aOR = 1.413, 95% CI 0.967–2.062, *p* = 0.074) as well as caregiver age. CLHIV residing in urban areas were 19.9% less likely to achieve undetectable viral load compared to their rural counterparts (aOR = 0.801, 95% CI 0.698–0.920, *p* = 0.002).

**Results from PSM.** After matching on several CLHIV characteristics, the study observed the following treatment effects for each of the outcomes (Table 6):

- **Viral rebound:** CLHIV who received at least one of the ACHIEVE project interventions were significantly less likely to experience viral rebound compared to those who had received none (β = − 0.047 (–0.072, –0.021), *p* < 0.001). This effect magnitude persisted in both average treatment effect (ATE) in the population and treatment effect on the treated (ATET).

- **Achieving undetectable viral load:** CLHIV who received at least one of the ACHIEVE project interventions were significantly more likely to achieve undetectable viral load at the follow-up compared to those who received none of the interventions (ATE: β = 0.10 (0.04, 0.16), *p* = 0.001; ATET: β = 0.11 (0.04, 0.17), *p* = 0.001).

## Discussion

This study explored whether ACHIEVE project interventions–economic strengthening through WORTH Yetu groups, linkage to teen/paediatric clubs, and health insurance–affected two clinical outcomes: (1) viral rebound and (2) achieving undetectable viral load, among CLHIV receiving ART in Tanzania. Overall findings revealed that CLHIV with undetectable viral load at baseline (Group 1) were at risk of viral rebound over time, while those with detectable viral load at baseline (Group 2) were more likely to achieve undetectable levels at follow-up, consistent with findings by Mutagonda et al [39]. Significant and positive associations were found between the project interventions and each outcome. Specifically, CLHIV

**Table 6. Propensity score matching (PSM) analysis of the average treatment effect (ATE), and average treatment effect on the treated (ATET) of ACHIEVE project interventions on viral rebound, and achieving undetectable viral load at follow-up among CLHIV receiving antiretroviral therapy in Tanzania.**

|  | Coefficient (β) | Lower 95% confidence limit | Upper 95% confidence limit | *p*-value |
|---|---|---|---|---|
| **Outcome 1: Experiencing viral rebound at follow-up (n = 21,448)** | | | | |
| Received ≥1 project intervention vs. zero (ATE) | –0.047 | –0.072 | –0.021 | < 0.001 |
| Received ≥1 project intervention vs. zero (ATET) | –0.048 | –0.074 | –0.021 | < 0.001 |
| **Outcome 2: Achieving undetectable viral load at follow-up (n = 4,809)** | | | | |
| Received ≥1 project intervention vs. zero (ATE) | 0.103 | 0.044 | 0.162 | 0.001 |
| Received ≥1 project intervention vs. zero (ATET) | 0.106 | 0.045 | 0.168 | 0.001 |

Factors matched on in PSM: ART regimen, Duration (in months) in the ACHIEVE project, CLHIV sex, CLHIV age, caregiver sex, caregiver age, caregiver education, CLHIV school enrolment status, level of household hunger, place of residence, family size, and whether the CLHIV changed ART regimen in the last 6 months. ATE: average treatment effect in the population; ATET: average treatment effect on the treated.

who received one or more of the three interventions were significantly less likely to experience viral rebound and more likely to achieve an undetectable viral load at follow-up compared to those who did not receive any of the interventions. Looking at each intervention individually, linkage to teen/paediatric clubs and health insurance were significantly associated with both outcomes, while WORTH Yetu was not. However, the combined effect of the three interventions on reducing viral rebound and achieving an undetectable viral load was statistically significant and much greater than the effect of each intervention on its own. This implies a synergistic effect, where the interactions among the interventions enhanced their overall effectiveness, suggesting that implementing the interventions together is more beneficial and offers greater effectiveness than implementing each intervention in isolation.

## Associations and impacts of the interventions on viral load

The reduction in the likelihood of viral rebound and the increased likelihood of achieving undetectable viral load at the follow-up among the CLHIV varied significantly based on the number of interventions received, demonstrating their cumulative impacts on both outcomes. This was such that the greater the number of interventions, the lower the likelihood of viral rebound among Group 1, and the higher the likelihood of achieving undetectable viral load among Group 2. CLHIV who received all three interventions experienced the largest reduction in the likelihood of viral rebound by 33.6%, followed closely by a 32.6% reduction for those who received any two interventions, and the lowest reduction of 17.4% for those who received one intervention. The three interventions had unequal effects on the likelihood of experiencing viral rebound: health insurance had the biggest and statistically significant reduction of 23.8%, followed by a statistically significant decline of 16.4% for linkage to teen/paediatric clubs, and the lowest but statistically insignificant decrease of 4.7% for the WORTH Yetu economic strengthening intervention.

Also, CLHIV who participated in the interventions were more likely to achieve undetectable viral load than those who did not. These impacts were largest, at 43.2% and 41.3%, for those who received two or three interventions, respectively, and lowest at 25.9% for those who received only one intervention. Again, the interventions had varying impacts, with teen/paediatric clubs being associated with the highest likelihood (26.7%) of achieving undetectable status, followed by health insurance at 23.5%. The WORTH Yetu economic strengthening intervention did not have a statistically significant effect on undetectable viral load status.

Although the economic strengthening intervention by itself was not statistically significant, it contributed to a synergistic effect: CLHIV who received all three interventions showed greater gains in both outcomes than those who received two, those who received two showed greater gains than those who received one, and one showed greater gains than those who received none.

Beyond the observed associations, the PSM analysis indicated that the interventions had positive and statistically significant impacts on both outcomes. There are several possible mechanisms by which the interventions could have contributed to the reduced likelihood of viral rebound and achieving undetectable viral load among CLHIV. First, economic strengthening can greatly enhance adherence to ART by reducing financial barriers to accessing healthcare and purchasing nutritious foods, which is essential for the effectiveness of ART. Previous studies have established a positive correlation between economic strengthening and food security [26], and between food security and adherence to ART among PLHIV [40,41]. In this study, caregivers may have improved their ability to afford transportation costs to clinics and maintain a healthy lifestyle that supports ART adherence, consequently leading to undetectable viral load.

Second, the reduced likelihood of viral rebound among CLHIV may be attributed to their participation in teen/paediatric clubs by way of the social support, education, and a sense of community provided through the clubs. These factors can enhance ART adherence among adolescents by addressing stigma and isolation, and by providing peer support. Social support networks are critical for adherence, especially among teenagers who may face unique challenges related to HIV stigma and treatment fatigue. These clubs can improve mental health and adherence to treatment plans [42], consequently improving HIV treatment outcomes among the CLHIV.

 

Third, health insurance is known to enhance healthcare access, promote health-seeking behaviour, and lower out-of-pocket medical expenses [43,44]. While HIV medical costs are provided to PLHIV for free in Tanzania, health insurance coverage for CLHIV and their families likely enabled caregivers to redirect saved funds towards buying nutritious foods and other necessities and covering transport costs to health facilities, thereby improving ART adherence and achieving sustained viral suppression. Also, improved caregiver health from better access to services through health insurance can enhance their ability to support their CLHIV's health.

Overall, the findings suggest that participation in these interventions supports CLHIV in achieving and maintaining undetectable viral loads, providing evidence to guide policy and resource allocation for scaling effective programs. Further research is needed to examine long-term effects and identify additional factors influencing clinical outcomes, to inform targeted interventions. Future studies could also assess cost-effectiveness of such programs, strengthen government ownership, and consider the broader context of HIV programming, including community-level investments, to ensure interventions reach all CLHIV, particularly those who are underserved. Given the current donor funding uncertainties, the government should develop robust mechanisms to enhance the sustainability of these community-level interventions, which are already demonstrating beneficial outcomes when they complement clinical services. Ensuring that such interventions are fully integrated into existing health systems will help safeguard the health, social, and emotional well-being of CLHIV and other PLHIV in Tanzania and similar settings elsewhere.

### Effects of other factors on viral load

Both outcomes of this study were significantly associated with several factors other than the ACHIEVE project interventions, including ART regimens, whereby CLHIV on regimens other than DTG-based regimens were more likely to experience viral rebound and less likely to achieve undetectable viral load at follow-up. Considering the known unmatched effectiveness of DTG and the low chances of developing DTG drug resistance [45–47], the regimen should be made more accessible to all eligible CLHIV, while additional support should be offered to address the unique needs of non-DTG users to optimize treatment outcomes and improve their quality of life.

Another factor was the caregiver's age, which was positively associated with both outcomes among the CLHIV. Specifically, CLHIV whose caregivers were in the age groups 30–39, 40–49, and 50–59 years were less likely to experience viral rebound; and those whose caregivers were in the age groups 40–49, and 50–59 years more likely to achieve undetectable viral load compared to CLHIV who had the youngest caregivers in the age group of 18–29 years. This pattern persisted for CLHIV with the oldest caregivers (60+ years), but it was not statistically significant for either outcome. There are at least two possible explanations for this observation. First, as caregivers age, they can improve in skills and economic stability, enabling them to offer stronger care and support to CLHIV, leading to better treatment outcomes. However, CLHIV with older caregivers (60+) were not significantly different from those with the youngest caregivers in both outcomes, possibly because the oldest caregivers are less productive and not active enough to adequately support the treatment of the CLHIV. Second, older caregivers may be more likely to engage in lighter income-generating activities (IGAs) closer to their homes and closely monitor CLHIV regarding medication adherence and food consumption. On the other hand, younger caregivers (aged 18–29 years) may be likely to be engaged in more intensive IGAs and often away from home as well as have low caregiving experience/skills, hence low support to their CLHIV.

Place of residence was another important factor, with CLHIV living in urban areas being less likely than their rural counterparts to experience positive outcomes for both viral rebound and undetectable viral load at the follow-up. Although urban settings are often assumed to offer greater access to services and opportunities, urban areas also tend to have higher costs of living, which can pose financial challenges for families of CLHIV in meeting basic needs such as affording nutritious foods and transportation to healthcare facilities. For families that are already poor and vulnerable, these challenges may be exacerbated by the urban paradox [48], whereby the presumed urban advantage coexists with pronounced inequalities that leave disadvantaged households, such as those of CLHIV, at risk of being left behind. Additionally, as

rural residents tend to be more communal than their urban counterparts, this may have played an important role in better treatment outcomes in rural areas. Studies have found that both caregiver emotional support and child social support are more common in rural areas compared to urban areas [49]. Place of residence has been associated with several health outcomes, including clinical care and health behaviours [50]. While urban CLHIV may require more context-specific support to address their needs and challenges, more research is needed to understand this association.

We observed in this study that the higher the caregiver's education level the higher the likelihood of viral rebound among the CLHIV. It could be that due to employment or other work-related commitments in which educated caregivers may be engaged, they may lack adequate time to consistently monitor and provide the required care and support for the CLHIV to achieve the desirable treatment outcomes. Another possibility could be that educated caregivers may assume that they possess sufficient knowledge to manage the child's HIV without seeking appropriate guidance or strictly adhering to medical advice which could lead to complacency in adhering to treatment regimens, monitoring the child's health status, or seeking timely medical attention, thereby increasing the risk of viral rebound. Therefore, CLHIV whose caregivers had primary and secondary or more education may be at higher risk of viral rebound or unlikely to remain undetectable, thus a need to target them with additional support for better and sustained HIV treatment outcomes. This finding requires further research to explain the underlying mechanisms clearly.

Finally, the finding that CLHIV who switched regimens in the last six months were more likely to experience viral rebound compared to those who did not could be due to reasons such as drug resistance or adherence issues. CLHIV who switched regimens may have developed resistance from the previous regimen and were possibly more likely to experience resistance to the new drug, hence at higher risk of experiencing viral rebound. Also, children may have varying immune responses to different ART regimens. Some regimens may not effectively boost the child's immune system, leaving them vulnerable to viral rebound after switching to a new regimen that is not well-suited to their immune needs.

Overall, this study showed that the interventions were associated with, and positively impacted, both viral rebound and undetectable viral load among CLHIV in Tanzania, highlighting the potential effectiveness of the interventions in achieving and maintaining viral suppression. This underlines the significance of the interventions in influencing clinical outcomes, highlighting the need to supplement clinical care and treatment for CLHIV with appropriate social support services to achieve and maintain desirable outcomes.

## Conclusion

Although 13.1% of CLHIV initially undetectable experienced viral load rebound at the follow-up, the ACHIEVE project interventions were significantly associated with decreased odds of viral rebound. Additional support may be needed, especially by those at an elevated risk of rebound, including those on regimens different from DTG, recently changing ART regimens, residing in urban areas, and having caregivers with primary, secondary, or higher education levels. Among CLHIV who initially had a detectable viral load, 70.9% achieved undetectable status at the follow-up, and the three ACHIEVE project interventions were significantly associated with a higher likelihood of achieving this outcome. The combined effect of the three interventions on reducing viral rebound and achieving an undetectable viral load was significantly greater than the impact of each intervention in isolation, suggesting a synergistic effect that enhances overall effectiveness. More targeted support is crucial for CLHIV at higher risk of persisting with a detectable viral load, particularly those residing in urban areas and those under the care of younger caregivers.

This study uncovers clear evidence that a holistic approach to CLHIV care, which integrates clinical services with community-based social support, is vital for achieving and maintaining desirable outcomes for CLHIV. This approach is instrumental in promoting both the health and social well-being of CLHIV.

## Supporting information

**S1 File. Supplemetary Tables S1 to S5. S1 Table.** Coverage of ACHIEVE project services among CLHIV as of July 15th, 2023. **S2 Table.** Factors associated with viral rebound at follow-up among 21,448 CLHIV who had undetectable viral load at baseline in Tanzania (ACHIEVE project interventions analysed as separate variables). **S3 Table.** Factors

associated with undetectable viral load at follow-up among 4,809 CLHIV who had detectable viral load at baseline in Tanzania (ACHIEVE project interventions analysed as separate variables). **S4 Table.** Factors associated with viral rebound at follow-up among 21,448 CLHIV who had undetectable viral load at baseline in Tanzania (ACHIEVE project interventions reduced into a single binary variable). **S5 Table.** Factors associated with undetectable viral load at follow-up among 4,809 CLHIV who had detectable viral load at baseline in Tanzania (ACHIEVE project interventions reduced into a single binary variable).

(ZIP)

## Acknowledgments

A version of this manuscript, based on Group 1 respondents, was presented as a poster (WEPEC296) at AIDS 2024, the 25th International AIDS Conference, held in Munich, Germany, from July 22–26, 2024. Further details about the presentation can be found at: https://programme.aids2024.org/Abstract/Abstract/?abstractid=3356. Also, another abstract based on Group 2 respondents was presented as a poster (TUPE040) at the HIVR4P 2024, the 5th HIV Research for Prevention Conference that took place in Lima, Peru, October 6–10, 2024.

## Author contributions

**Conceptualization:** Amon Exavery, Daisy Kisyombe, Alison Koler, John Charles, Asheri Barankena, Tom Ventimiglia, Levina Kikoyo.

**Data curation:** John Charles.

**Formal analysis:** Amon Exavery, Daisy Kisyombe, Alison Koler.

**Funding acquisition:** Jennifer Mulik, Gloria Sangiwa, Levina Kikoyo.

**Methodology:** Amon Exavery, Daisy Kisyombe, Alison Koler, John Charles, Nemes Mallya, Erica Kuhlik, Asheri Barankena, Rose Fovo, Godfrey Martin Mubyazi, Tom Ventimiglia.

**Resources:** Jennifer Mulik.

**Software:** Amon Exavery.

**Supervision:** Jennifer Mulik, Gloria Sangiwa, Levina Kikoyo.

**Validation:** John Charles.

**Visualization:** Amon Exavery.

**Writing – original draft:** Amon Exavery.

**Writing – review & editing:** Amon Exavery, Daisy Kisyombe, Alison Koler, Nemes Mallya, Erica Kuhlik, Asheri Barankena, Rose Fovo, Godfrey Martin Mubyazi, Tom Ventimiglia, Jennifer Mulik, Gloria Sangiwa, Levina Kikoyo.

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
