## [Decision Letter · Decision Letter 0]

14 Dec 2025

PONE-D-25-21420Evaluating the effect of community-based programs on viral load among HIV-positive orphaned and vulnerable children on antiretroviral treatment: findings from the ACHIEVE project in TanzaniaPLOS One

Dear Dr. Exavery,

Thank you for submitting your manuscript to PLOS ONE. After careful consideration, we feel that it has merit but does not fully meet PLOS ONE’s publication criteria as it currently stands. Therefore, we invite you to submit a revised version of the manuscript that addresses the points raised during the review process.

We look forward to receiving your revised manuscript.

Kind regards,

Sarah Nanzigu, Ph.D.,MSc.,MBchB

Academic Editor

PLOS One

Journal Requirements:

This paper is based on data from the ACHIEVE project in Tanzania, 2020 - 2026. The project is supported by PEPFAR through the USAID.

This study is part of the ACHIEVE project, funded by PEPFAR through USAID. A version of this manuscript, based on Group 1 respondents, was presented as a poster (WEPEC296) at AIDS 2024, the 25th International AIDS Conference, held in Munich, Germany, from July 22-26, 2024. Further details about the presentation can be found at: https://programme.aids2024.org/Abstract/Abstract/?abstractid=3356. Also, another abstract based on Group 2 respondents was presented as a poster (TUPE040) at the HIVR4P 2024, the 5th HIV Research for Prevention Conference that took place in Lima, Peru, October 6-10, 2024.

This paper is based on data from the ACHIEVE project in Tanzania, 2020 - 2026. The project is supported by PEPFAR through the USAID.

I have read the journal's policy and the authors of this manuscript have the following competing interests: One author, Amon Exavery, was awarded a travel scholarship by the International AIDS Society (IAS) to attend and present an abstract based on Group 2 respondents at the HIVR4P 2024, the 5th HIV Research for Prevention Conference, held in Lima, Peru, October 6–10, 2024. The rest of the authors declared that no competing interests exist.

6. In the online submission form, you indicated that deidentified participant data will be available upon reasonable request. Any additional requests for data access or analysis must receive prior approval from Pact. Such requests should be directed to Dr. Levina Kikoyo at lkikoyo@pactworld.org.

**Additional Editor Comments:**

The manuscript needs serious attention to eliminate inconsistencies and improve clarity. Some of the inconsistencies have been presented under comments from reviewer 1.

In addition, kind attend to the following:

1. Reconcile names of community program in the your letter to the editor ( currently stated as 'WORTH Yetu economic strengthening') with the name presented in your manuscript title (ACHIEVE).

2. Your study title and abstract seem to represent just a subset of the entire study. The first two study objectives presented in the main body (lines 111- 118) are not captured in your title and abstract. Moreover, the evaluation of 'the impact of ACHIEVE' as an objective is so marginalized that a reader would consider it a secondary objective. I would encourage authors to rethink their study report and line it with their original study aims.

3. Just to emphasize a comment by reviewer 1, of the need for authors to clearly state the inclusion/ exclusion criteria, the report seems to mix up 'period on ART' and 'duration under ACHIEVE'. While lines 158-160 states that CLHIV who had been on ART for at least six months and had at least two clinically confirmed viral load test results spaced at least six months apart between July 2021 and July 2023', the table of baseline characteristics on page 12 emphasizes on duration on the ACHIEVE project', and goes ahead to include participants' duration less than 6 months. A reader would wonder if this group was eligible for the study. The authors therefore need to clearly state the inclusion/ exclusion criteria, and consistently apply the relevant variables in the analysis.

4. The tables of results are overcrowded with a lot information. Authors should consider limiting themselves to data most relevant to their report. Just to cite examples: Is it of much relevant to your report that caregiver's age be categorized into 5 groups and then family size into 3 groups?

Reviewers' comments:

Reviewer's Responses to Questions

**Comments to the Author**

1. Is the manuscript technically sound, and do the data support the conclusions?

Reviewer #1: Yes

Reviewer #2: Yes

2. Has the statistical analysis been performed appropriately and rigorously? 

Reviewer #1: Yes

Reviewer #2: Yes

3. Have the authors made all data underlying the findings in their manuscript fully available?

Reviewer #1: Yes

Reviewer #2: Yes

4. Is the manuscript presented in an intelligible fashion and written in standard English?

Reviewer #1: Yes

Reviewer #2: Yes

5. Review Comments to the Author

Reviewer #1: This manuscript is technically sound and result of their findings supports the conclusion. The presented data, the methodology and the details of the analysis shows that they used standard data analysis software and were able to different variables and possible factors that could affect the outcome. However, I observe that the authors needs few major revisions on some of the Tables as I have highlighted in the manuscript to aid the readers to appreaciate the result of their findings in a more simplified and clear fashion without much ambiguity.

I have made comments on the body of the work as they would find in the attached document.

Reviewer #2: Thank you for the opportunity to review this interesting article on community base programming and the influence of three categories of interventions on viral load among children receiving ARVs.

While the article is potentially of interest to a subset of readers who are programming with certain pediatric populations in mind, the interventions that were the focus of this report are not especially innovative, and the specifics of these interventions is unclear. Even if one is to pursue similar programming based on these findings, it would be nearly impossible to reproduce the activities leading to the reported impact.

Importantly, there is very little discussion of sustainability, cost-effectiveness, government ownership and the current context of HIV programming (including community-level investments) as of this review in late 2025.

In fact, for those following the future of programming and related budgets in Tanzania and other countries in the region, the manuscript comes off as disjointed from reality and largely naive to the latest driving programmatic political factors that will ultimately determine the success of such programs, and their relevance.

Indeed, the funding agency that is behind these investments no longer exists, and the upcoming five years will look nothing like the previous 20. If one is to look for similar studies over the last decade, s/he will find several, including from Tanzania.*** While the Konga community-based intervention model seems to be implemented by the National Council of People Living with HIV (NACOPHA), the relationship of current work with he many years preceding would offer important context. To imporve the value of this manuscript. the report needs to include much more information placing this study in current context.

With a re-write to update the paper in accordance with current context and placing these interventions in context (and providing more information on how the activities are executed so that they can be replicated), the Editor may consider accepting the article. Of note, much of the existing literature focuses on adolescents and youth, and CLHIV data is sparse.

A rooted paper informing the future of practical community programming among this young and highly vulnerable population would be welcome.

---

***

Ferrand RA et al. 2017. Community-based caregiver support to reduce virological failure in children and adolescents with HIV (ZENITH): Randomised controlled trial. Lancet Child Adolesc Health.

Chikwari CD et al. 2018. Community health worker support for older children and adolescents with HIV: Process evaluation of the ZENITH trial. Implementation Science.

Willis N et al. 2019. Effectiveness of Community Adolescent Treatment Supporters (CATS) on linkage, retention, and adherence among adolescents with HIV: Randomised trial. BMC Public Health.

Mavhu W et al. 2020. Differentiated peer-led service delivery for adolescents with HIV (Zvandiri): Cluster randomised trial. Lancet Global Health.

Ndhlovu CE et al. 2021. Peer-support intervention to improve ART outcomes among adolescents and young adults with virologic failure: Randomised controlled trial. AIDS Research & Therapy.

Simms V et al. 2022. Peer-led Problem Discussion Therapy for adolescents living with HIV: Cluster-randomised trial. PLOS Medicine.

Dhlamini N et al. 2019. Peer-led HIV care continuum outcomes in the Zvandiri programme. Global Health: Science & Practice.

Mageda K et al. 2023. Effectiveness of a community-based Konga model to improve viral suppression in children with HIV: Cluster-randomised clinical trial. BMC Public Health.

Mongi J et al. 2023. Community-based differentiated service delivery for paediatric HIV in Tanzania: Improved viral suppression and retention. HIV Pediatrics Conference Proceedings.

Msomi N et al. 2023. Determinants of viral load suppression among orphaned and vulnerable children receiving community-based caseworker support. Frontiers in Public Health.

Kahema L et al. 2022. Determinants of viral load non-suppression among HIV-positive children and adolescents in community treatment settings in Tanzania. Bulletin of the National Research Centre.

Grimwood A et al. 2012. Community adherence support and treatment outcomes among children on ART: Multicentre cohort study. Journal of the International AIDS Society.

Machiha A et al. 2024/2025. Integrated community-based HIV and SRH services for youth (CHIEDZA): Cluster randomised trial. Nature Medicine.

6. PLOS authors have the option to publish the peer review history of their article (what does this mean?). If published, this will include your full peer review and any attached files.

Reviewer #1: **Yes:** Nnenna Assumpta EZEOKAFOR

Reviewer #2: No

---

## [Author Response · Author response to Decision Letter 1]

4 Feb 2026

A Response to Reviewers file has been uploaded, detailing how each comment was addressed and indicating the corresponding changes in the revised manuscript.

---

## [Decision Letter · Decision Letter 1]

26 Mar 2026

PONE-D-25-21420R1Evaluating the effects of community-based programs on viral rebound and viral suppression among HIV-positive orphaned and vulnerable children receiving antiretroviral treatment: findings from the ACHIEVE project in TanzaniaPLOS One

Dear Dr. Exavery,

Thank you for submitting your manuscript to PLOS ONE. After careful consideration, we feel that it has merit but does not fully meet PLOS ONE’s publication criteria as it currently stands. Therefore, we invite you to submit a revised version of the manuscript that addresses the points raised during the review process.

We look forward to receiving your revised manuscript.

Kind regards,

Tacilta Nhampossa

Academic Editor

PLOS One

**Journal Requirements:**

1, If the reviewer comments include a recommendation to cite specific previously published works, please review and evaluate these publications to determine whether they are relevant and should be cited. There is no requirement to cite these works unless the editor has indicated otherwise.

Reviewers' comments:

Reviewer's Responses to Questions

**Comments to the Author**

1. If the authors have adequately addressed your comments raised in a previous round of review and you feel that this manuscript is now acceptable for publication, you may indicate that here to bypass the “Comments to the Author” section, enter your conflict of interest statement in the “Confidential to Editor” section, and submit your "Accept" recommendation.

Reviewer #1: All comments have been addressed

Reviewer #2: All comments have been addressed

Reviewer #3: All comments have been addressed

2. Is the manuscript technically sound, and do the data support the conclusions?

Reviewer #1: Yes

Reviewer #2: Yes

Reviewer #3: Yes

3. Has the statistical analysis been performed appropriately and rigorously? 

Reviewer #1: Yes

Reviewer #2: Yes

Reviewer #3: Yes

4. Have the authors made all data underlying the findings in their manuscript fully available?

Reviewer #1: Yes

Reviewer #2: Yes

Reviewer #3: Yes

5. Is the manuscript presented in an intelligible fashion and written in standard English?

Reviewer #1: Yes

Reviewer #2: Yes

Reviewer #3: Yes

6. Review Comments to the Author

Reviewer #1: The authors have adequately addressed the comments raised in the previous round of review; however, there are a few minor revisions I ave observed that should be corrected before publication. I have highlighted the comments in the attached document

Reviewer #2: This reviewer appreciates the thoughtful responses to all elements in the initial review. Thank you for the opportunity to provide feedback.

Reviewer #3: One main comment about the title (line 5 of the PDF version: Use of the term "HIV-positive": this terminology is not appropriate per UNAIDS guidance. Replace with “living with HIV”.

7. PLOS authors have the option to publish the peer review history of their article (what does this mean?). If published, this will include your full peer review and any attached files.

Reviewer #1: No

Reviewer #2: No

Reviewer #3: No

---

## [Author Response · Author response to Decision Letter 2]

5 Apr 2026

We have uploaded a file titled "Response to Reviewers", which contains our responses to each of the reviewer and editor comments.

---

## [Decision Letter · Decision Letter 2]

26 Apr 2026

Evaluating the effects of community-based programs on viral rebound and viral suppression among HIV-positive orphaned and vulnerable children receiving antiretroviral treatment: findings from the ACHIEVE project in Tanzania

PONE-D-25-21420R2

Dear Dr. Exavery,

We’re pleased to inform you that your manuscript has been judged scientifically suitable for publication and will be formally accepted for publication once it meets all outstanding technical requirements.

Kind regards,

Tacilta Nhampossa

Academic Editor

PLOS One

Additional Editor Comments (optional):

Reviewers' comments:

Reviewer's Responses to Questions

**Comments to the Author**

1. If the authors have adequately addressed your comments raised in a previous round of review and you feel that this manuscript is now acceptable for publication, you may indicate that here to bypass the “Comments to the Author” section, enter your conflict of interest statement in the “Confidential to Editor” section, and submit your "Accept" recommendation.

Reviewer #1: All comments have been addressed

2. Is the manuscript technically sound, and do the data support the conclusions?

Reviewer #1: Yes

3. Has the statistical analysis been performed appropriately and rigorously? 

Reviewer #1: Yes

4. Have the authors made all data underlying the findings in their manuscript fully available?

Reviewer #1: Yes

5. Is the manuscript presented in an intelligible fashion and written in standard English?

Reviewer #1: Yes

6. Review Comments to the Author

Reviewer #1: I appreaciate the opportunity to review this piece of work. The authors have done a great job in developing the manuscript and contributing to the body of knoweldege that can help guide future programming especially among the CLHIV with opportunity for further research for strategies for scaling impact in the long term. They have also been able to address all comments in the second round of the review.

7. PLOS authors have the option to publish the peer review history of their article (what does this mean?). If published, this will include your full peer review and any attached files.

Reviewer #1: No

---

## [Editor Report · Acceptance letter]

PONE-D-25-21420R2

PLOS One

Dear Dr. Exavery,

I'm pleased to inform you that your manuscript has been deemed suitable for publication in PLOS One. Congratulations! Your manuscript is now being handed over to our production team.

Kind regards,

on behalf of

Dr. Tacilta Nhampossa

Academic Editor

PLOS One